# Integration of primary contact physiotherapists in the emergency department for individuals presenting with minor musculoskeletal disorders: Protocol for an economic evaluation

Rose Gagnon[1,2], Luc J. Hébert[1,2,3], Jason R. Guertin[4,5], Simon Berthelot[5,6,7], François Desmeules[8,9], Kadija Perreault[1,2]*

1 Centre Interdisciplinaire de Recherche en Réadaptation et Intégration Sociale, Centre Intégré Universitaire de Santé et de Services Sociaux de la Capitale-Nationale, Quebec, Quebec, Canada, 2 Department of Rehabilitation, Faculty of Medicine, Université Laval, Quebec, Quebec, Canada, 3 Department of Radiology and Nuclear Medicine, Faculty of Medicine, Université Laval, Quebec, Quebec, Canada, 4 Department of Social and Preventive Medicine, Faculty of Medicine, Université Laval, Quebec, Quebec, Canada, 5 Axe Santé des Populations et Pratiques Optimales en Santé, Centre de Recherche du Centre Hospitalier Universitaire de Québec, Université Laval, Quebec, Quebec, Canada, 6 Department of Family Medicine and Emergency Medicine, Faculty of Medicine, Université Laval, Quebec, Quebec, Canada, 7 Centre Hospitalier Universitaire de Québec, Université Laval, Quebec, Quebec, Canada, 8 School of Rehabilitation, Faculty of Medicine, Université de Montréal, Montreal, Quebec, Canada, 9 Orthopaedic Clinical Research Unit, Maisonneuve-Rosemont Hospital Research Centre, Centre Intégré Universitaire de Santé et de Services Sociaux de l'Est-de-l'Île-de-Montréal, Montreal, Quebec, Canada

* kadija.perreault@fmed.ulaval.ca

**Data Availability Statement:** No datasets were generated or analyzed during the current study. All

## Abstract

### Objectives

1) To compare the average cost of an emergency department (ED) visit for various minor musculoskeletal disorders between two models of care (physiotherapist and ED physician or ED physician alone); 2) To evaluate the incremental cost-effectiveness ratio (ICER) of these two models of care over a 3-month period post-initial visit; and 3) To estimate the ICER of three ED models of care (physiotherapist and ED physician, ED physician alone, physiotherapist alone) over a two-year period.

### Methods

Obj.1: The costs incurred by participants in the two groups during their ED visit will be calculated using the Time-Driven Activity-Based Costing (TDABC) method. These costs will be compared using generalized linear models. Obj. 2: The ICER of the two models will be evaluated over three months via a cost-utility analysis that will combine costs and effectiveness data (quality-adjusted life years) using both Health system and Societal perspectives (patient + health system costs). Obj. 3: The 2-year ICER of the three above-mentioned models will be estimated using a mathematical model including a decision tree (0–3 months post-visit) and a Markov model (3–24 months post-visit), also using both Health system and

relevant data from this study will be made available upon study completion.

**Funding:** Part of the data that will be used in this study were acquired during a randomized clinical trial that was supported by the CHU de Québec – Université Laval, subsidies from LJH and KP, and a clinical research scholarship awarded to one of the CHU de Québec collaborators by the Fondation du CHU de Québec for the multidisciplinary council of the CHU de Québec – Université Laval. RG received scholarships from the Canadian Institute of Health Research (CIHR #491913; https://cihr-irsc.gc.ca/e/193.html), the Fonds de recherche du Québec – Santé (FRQ-S; https://frq.gouv.qc.ca/sante/), the Unité de Soutien SSA – Québec (https://ssaquebec.ca/), the Ordre professionnel de la physiothérapie du Québec (OPPQ; https://oppq.qc.ca/), the Centre interdisciplinaire de recherche en réadaptation et intégration sociale (Cirris; https://www.cirris.ulaval.ca/) and Université Laval (https://www.fmed.ulaval.ca/). There was no additional external funding received for this study. JRG, SB and FD are FRQ-S Research Scholars. The funders had no role in study design, data collection and analysis, decision to publish, or preparation of the manuscript.

**Competing interests:** The authors have declared that no competing interests exist.

Societal perspectives. Data to answer the three objectives will come from data collected during a randomized clinical trial (n = 78, CHU de Québec)which will be supplemented with data obtained via some of the CHU de Québec administrative databases (nominative data; SIURGE (ED management software), Cristal-Net (patient electronic record), and the ED's pharmacy transactions directory; administrative data: drug costs repository), the literature, and public cost repositories.

## Conclusion

This study will help to determine which model of care is most efficient for the management of individuals who come to the ED with minor musculoskeletal disorders. The increased involvement of various health professionals in the management of patients in the ED paves the way for the development of new avenues of practice and more efficient organization of services.

## Introduction

The emergency department (ED) serves as the main gateway and the preferred resource when primary care services are not available, for example in cases of lack of affiliation with a primary care source or inability to see a physician within a reasonable time frame [1–5]. Although pain conditions for which patients decide to go to the ED are varied, they are oftentimes related to a musculoskeletal disorder (MSKD) [6–8].

According to the World Health Organization, MSKDs are characterized by "pain (often persistent) and limitations in mobility, dexterity and general functioning" [9]. MSKDs can affect joints, bones, muscles, spine and multiple regions of the body [9, 10]. The prevalence of these disorders is reported to be significantly higher in women, older people and people with low socio-economic status [11–15]. When they do not receive timely and appropriate care, people with MSKDs tend to make greater use of health care services and resources [16–22]. MSKDs account for up to 12.6% of a country's total health care costs each year [15] and this figure is expected to rise with the increase in obesity, physical inactivity and the aging of the population [11, 23]. It is therefore essential to study the costs and clinical effectiveness of interventions aimed at managing MSKDs in order to choose the most efficient ones, including in the ED.

Various models of care have been implemented in the ED and studied in recent years to optimize the management of people presenting with MSKDs. These models of care aim to optimize the flow of patients to and in the ED in three distinct phases: "input" (i.e., flow of patients deciding to come to the ED), "throughput" (i.e., flow of patients while in the ED), and "output" (i.e., flow of patients upon discharge from the ED) [24]. Such models of care include for instance fast-track corridors for patients with minor injuries or rapid assessment teams [25]. Some models include the addition of ED nurse practitioners and a variety of health professionals with a usual or extended scope of practice, such as the primary contact physiotherapist or advanced practice physiotherapist [25].

The addition of primary contact physiotherapists in the ED is an emerging model of care that aims to optimize patient flow while in the ED [25]. Several studies conducted in recent years have shown that this model of care is associated with reduced time waited before receiving care, and reduced length of stay in the ED, as well as fewer unnecessary consultations with various health professionals, and less prescriptions of imaging tests and medication, including

opioids, and over-the-counter medication [8, 26–29]. In addition, this model of care was associated with fewer repeat visits to the ED for a similar condition for up to one month after the initial ED visit [29]. Thus, management by a primary contact physiotherapist appears to be associated with decreased service and resource use, both at the ED and up to several weeks later. However, very few studies having investigated primary contact physiotherapist care in the ED have looked at its cost-effectiveness.

Indeed, despite evidence of clinical benefits associated with the presence of a primary contact physiotherapist in the ED (effectiveness), scientific evidence remains rather scarce regarding the cost-effectiveness of this model of care. Two studies conducted in primary care settings (primary care clinic and private clinic) report that primary contact physiotherapist management is associated with a slight increase in health-related quality of life and a decrease in total costs compared to usual management by a family physician [30, 31]. In addition, early physiotherapy management was associated with a decrease in total MSKD-related costs for up to two years after initial management [17, 19, 32]. Two cost-minimization studies conducted in Great Britain looked specifically at the costs associated with the integration of a primary contact physiotherapist in the ED compared to usual management by an emergency physician. According to the study by Richardson et al. (2005, n = 766 patients with non-fracture MSKDs), the presence of a primary contact physiotherapist in the ED results in costs equivalent to usual management (emergency physician) [33]. Similarly, according to McClellan et al. (2013, n = 372 patients >16 years of age with a peripheral MSKD), management by a primary contact physiotherapist results in costs at least as high as usual management (emergency physician) [34]. Nevertheless, in addition to having been conducted exclusively in Great Britain several years ago, these two studies only measured the costs of the two models of care compared and not their effectiveness, the authors assuming that the two models compared were equivalent in terms of clinical effectiveness. These studies are thus not considered to be formal economic evaluation according to current guidelines, but rather a costing exercise, in that a cost-effectiveness analysis accounts for the uncertainty associated with the effects of the interventions being compared [35]. To our knowledge, no other study has examined the cost-effectiveness of primary contact physiotherapy in the ED. Furthermore, no study has assessed whether involving primary contact physiotherapists in the ED have a long-term impact on use of health system services and resources for persons with minor MSKDs. Consequently, further evidence is needed on the efficiency of integrating a primary contact physiotherapist in the ED compared to usual management by an emergency physician.

Therefore, the general objective of this project is to evaluate the efficiency of different models of care for the management of minor MSKDs in the ED. More specifically, the objectives are to:

1. Compare the average costs of an ED consultation and care for various MSKDs, according to two models of care:

   a. Usual management by an emergency physician

   b. Primary contact physiotherapist management + emergency physician management

2. Evaluate the incremental cost-effectiveness ratio (ICER), from both Health system and Societal perspectives, of these two ED models of care for the management of MSKDs over a three-month period post-initial ED visit.

3. Estimate the ICER between three ED models of care for MSKD management over a two-year period from both Health system and Societal perspectives:

   a. Usual management by an emergency physician

b. Primary contact physiotherapist management + emergency physician management

c. Primary contact physiotherapist management alone

## Materials and methods

### Study design and costing approaches

This study is composed of three distinct designs, one per objective. The costing approach used for Objective 1 will be Time-Driven Activity-Based Costing (TDABC), which involves determining the per-minute costs associated with each care process included in a care pathway by multiplying the cost per minute of each care process by its duration. Details on this costing approach and its application to the ED have been described by one of the authors elsewhere [36]. Objective 2 will be achieved through a cost-utility analysis approach in which health care costs at the ED visit and those reported at the 1- and 3-month follow-ups will be compiled and combined with the utility scores obtained at the same measurement times, from both Health system and Societal perspectives. Cost-utility analysis is favored in Canada since it uses a generic outcome measure allowing comparison of the health gains associated with several different interventions, such as different models of care [35].

The ICER between the three ED models of care for the management of MSKDs over a two-year period (Objective 3) will be estimated using a cost-effectiveness analysis via a hybrid mathematical model. This model will consist of a decision tree covering the period from the initial ED visit up to three months post-initial visit, and a Markov model starting three months post-initial ED visit and ending 24 months (two years) after the ED visit. The decision tree provides a simple and clear illustration of a patient's possible short-term care pathways following a new intervention [35, 37]. In addition to reporting the different interventions used, the decision tree also allows for the inclusion of adverse events following the initial intervention, such as a new ED visit for the same condition, and for repeating an intervention over time as needed (e.g., new visit in the ED a few days after the initial visit and then a new visit two months later for the same condition) [35, 37]. It also permits to determine the proportion of disability associated with each of the three ED models of care.

Several considerations guided the choice of the time horizon for the Markov model. First of all, to be considered chronic, a musculoskeletal disorder must be present for at least three months [38]. Moreover, approximately 30% of people presenting with MSKDs report pain and functional disability lasting more than 12 months after the onset of their condition. Furthermore, studies on MSKD care in primary care or the ED have had follow-up periods ranging from six to 24 months (e.g. [26, 39–42]). Thus, the Markov model will cover a 24-month period. It will include two-week cycles in order to capture the clinical evolution of the patients included.

### Population of interest

Inclusion and exclusion criteria for the population of interest are described in Box 1. The population targeted covers persons with a peripheral or spinal MSKD, a P3, P4 or P5 triage category at the ED (Canadian Triage and Acuity Scale [43]), and that are aged between 18 and 80. Having a major MSKD, red flag or associated unstable condition are criteria for exclusion.

### Data collection

Objectives 1, 2 and 3 will be achieved in part using data collected through a two-arm pilot pragmatic randomized clinical trial (RCT) conducted in the ED of the CHUL, one of the five

**Inclusion criteria**

- Disorder or pain of musculoskeletal origin peripheral or vertebral
- Aged between 18 and 80 years old
- P3, P4 or P5 Triage Category (classification from the Canadian Triage and Acuity Scale)
- Legally able to consent
- Able to understand French and respond to oral or written questionnaires
- Beneficiary of the *Régie de l'assurance-maladie du Québec*

**Exclusion criteria**

- Major MSKD (e.g., open fracture, unreduced dislocation, open wound)
- Red flag (e.g., progressive neurological deficits, infectious symptoms)
- Associated unstable condition (e.g., pulmonary, cardiac, digestive and/or psychiatric)
- Condition involving being already hospitalized during the recruitment period or having already been hospitalized for this same condition beforehand
- Coming from a long-term care centre

Box 1. Inclusion and exclusion criteria for the population of interest.

sites of the CHU de Québec—Université Laval (UL) (Quebec City, Canada) from September 2018 to March 2019 (n = 78). This trial aimed to compare the effects of management by a primary contact physiotherapist to usual care provided by an emergency physician for persons presenting with a minor MSKD on their clinical course (pain and pain interference) and the use of resources at ED discharge and after 1 and 3 months post-visit [29]. Two groups of participants were compared: one group managed by a primary contact physiotherapist and an emergency physician and one group managed by an emergency physician alone.

Data were collected at the initial ED visit and at the one- and three-month post-visit follow-ups. While more details of the data collection procedures can be found in our previous paper [29], any person presenting to the ED who met the inclusion and exclusion criteria was seen by a member of the research team who confirmed eligibility, obtained informed consent, and ensured completion of baseline questionnaires. The participant was then randomized to either study group: primary contact physiotherapist + emergency physician management or usual management by the emergency physician alone. After the ED visit was completed, participants were contacted at 1 and 3 months either by phone or email to complete post-visit follow-ups. The supplementary data needed to meet the three Objectives will come from some of the CHU de Québec—UL administrative databases (nominative data: SIURGE (ED management software), Cristal-Net (patient electronic record), and the CHUL's ED pharmacy transactions directory; administrative data: prescription and over-the-counter drug costs repository), the scientific literature, and public cost repositories (e.g., salaries of general practitioners and specialists, see https://www.msss.gouv.qc.ca/inc/documents/ministere/acces_info/seance-publique/etude-credits-2018-2019/Reponses-aux-questions-generales-et-particulieres-RAMQ.pdf; laboratory analysis costs, see https://publications.msss.gouv.qc.ca/msss/fichiers/2017/17-922-05W.pdf).

Of note, the study population for the model of care consisting of primary contact physiotherapist management and discharge from the ED was not observed at all during the pilot

pragmatic RCT. Consequently, this model of care will have to be modeled, and therefore cannot be studied in the context of Objectives 1 and 2, which are based on RCT data. All the parameters needed to represent this care model within Objective 3 (probabilities, costs, measures of effectiveness) will be taken from a literature review, an approach regularly used in economic evaluation [44]. However, the studies from which the metrics will be derived will need to have a sample that meets the same inclusion criteria as those presented in Box 1. Data extracted from the literature will be validated with members of the research team and with experts in the field of emergency medicine, MSKDs and rehabilitation if necessary during the construction of the hybrid model [35].

## Study outcomes

Primary outcomes used to measure the average cost of an ED visit (Objective 1) will be the costs of care processes and the time associated with each care process. This method of costing is routinely used by some members of the research team [36, 45, 46]. The costs related to ED management (medical and non-medical staff, imaging, medication, consumables, maintenance, etc.) were obtained via a formal request made by a member of the research team to the CHU de Québec–UL Finance Department. The time associated with each care process was calculated by a member of the research team using estimates provided by the CHUL medical and non-medical staff that were validated during an observation period in the ED [36, 45]. Uncertainty in measured times will allow the variability of ED costs to be considered, and to derive a distribution of possible costs at baseline for each participant.

As part of the cost-utility analysis (Objective 2) and hybrid mathematical model (Objective 3), the efficiency of the ED models of care will be assessed using an incremental cost-effectiveness ratio (ICER). The resulting ICER will be reported in terms of incremental cost per quality-adjusted life years (QALY) gained, between the models of care. The follow-up costs of each participant recorded via the self-administered follow-up questionnaires completed at 1 and 3 months during the pilot pragmatic RCT will be added to the distribution of ED visit costs for each individual obtained under Objective 1. Once all costs have been added, an average will be calculated to obtain an average cost per participant at 3 months, for each model of care (Objective 2 & 3 –decision tree). The questionnaires provided data on resources used by each participant during follow-up such as ED re-visits for the same condition, number of consultations with other health professionals in the public and private sector, imaging tests used, etc. The costs of all resources used in the public healthcare system will be drawn from public cost repositories data from the *Régie de l'assurance-maladie du Québec* (RAMQ) (costs related to the emergency physician and other physicians consulted, drugs, laboratory analysis, imaging tests) [47]. Hourly rates for the primary contact physiotherapist and other health professionals consulted (e.g., massage therapist, chiropractor, osteopath) will be derived from a search of the grey literature and will be based on rates in effect within the private healthcare system, since the majority of these professionals work within this system [47]. Costs related to the use of technical aids (e.g., cane, crutches, walker) will also come from a grey literature search. It should be noted that the salary data available from human resources for all healthcare professionals (physicians and allied health professionals) considered will not allow for variability in hourly costs. The only source of variability considered for the purpose of these analyses will be that related to the time required for ED processes. Utility scores were obtained at the initial visit [48] and at 1 and 3 months using the EQ-5D-5L, a generic standardized questionnaire designed to measure health status in an economic and clinical evaluation [49]. The EQ-5D-5L has been found to be reliable, valid, and sensitive to change [50, 51]. The difference in utility scores between the 3-month follow-up and baseline will be calculated for each participant

using area-under-the-curve analyses. Once the difference in utility scores will be calculated for each participant, the differences will be averaged to obtain the average gain or loss in utility scores per ED care model. The mean gain or loss in utility scores will then be transformed into QALYs. The efficiency values and the costs from 3 to 24 months required to run the Markov model (Objective 3) for the three models of care will be taken from the literature.

## Data analysis and interpretation of results

As part of Objective 1, a mapping of the care pathways encountered will be completed for each type of MSKD encountered in our study population (i.e., low back pain, neck pain, upper limb, lower limb) (Fig 1). The unit cost of each of the resources, consumables and indirect costs required in each process of care of the care pathway will be calculated and multiplied by the duration of each process to obtain the cost related to each process of care present in the care pathway. The costs associated with each process will be summed to obtain the total cost of the ED care pathway specific to each MSKD and each model of care (i.e., emergency physician or primary contact physiotherapist and emergency physician management). A generalized linear model with a Gamma distribution and log link will be used to test whether there is a significant difference in the costs of managing equivalent MSKDs between the two models of care [52, 53].

The decision tree (Objective 3) will be created to reflect the results of the RCT as closely as possible and will therefore include all possible interventions and services used by a participant following the initial visit to the ED for each model of care considered (Fig 2). The pruning of each terminal node containing less than 5% of the participants will be determined based on discussions with a panel of experts. The conditional probability of ending up in each of the remaining terminal nodes of the decision tree will be used to calculate the proportion of disability associated with each care model (Fig 2). The disability proportions obtained for each model of care will be used to determine the number of individuals in each state at entry in the Markov model (Fig 3). The Markov model will then be used to calculate the long-term costs and effectiveness over 2 years of each of the model of care based on the level of disability estimated in the decision tree [35, 37].

Both the cost-utility analysis (Objective 2) and the hybrid mathematical model (Objective 3) will be conducted from a Health system and a Societal perspective. Results obtained via the EQ-5D-5L at 1 and 3 months (Objective 2 & 3 –decision tree) will be converted to utility scores using the Canadian conversion algorithm developed by Xie et al. [54]. As the 3-month retention rate for the pilot pragmatic trial was 80% [29], some participants' data are missing (service and resource use, costs, utility scores). Missing data will be imputed using the Missing not at random (MNAR) multiple imputation method [55]. Patient characteristics believed to have

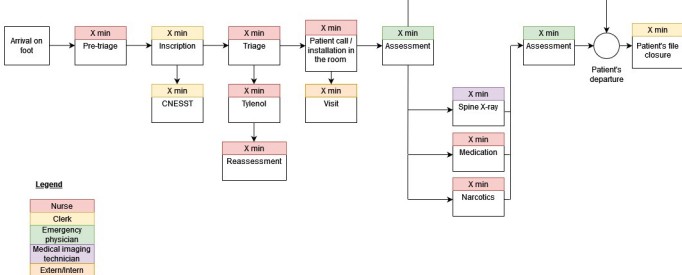

**Fig 1. Mapping of a hypothetical care pathway in the ED using the Time-Driven Activity-Based Costing.**

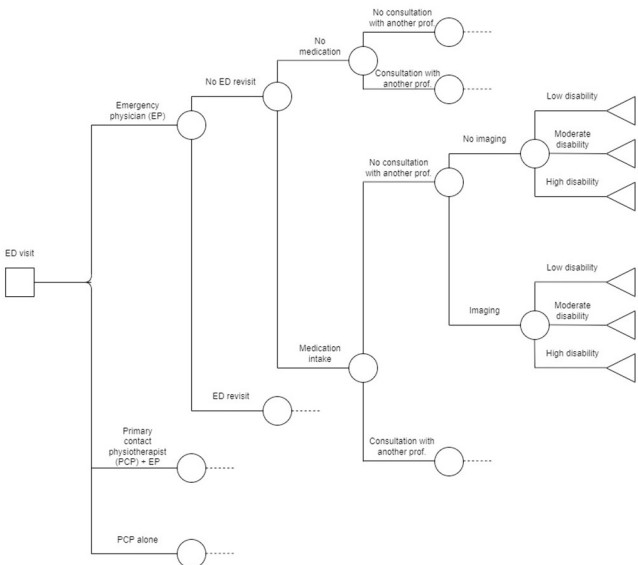

**Fig 2. Hypothetical decision tree covering the period from ED visit to three months post initial ED visit.**

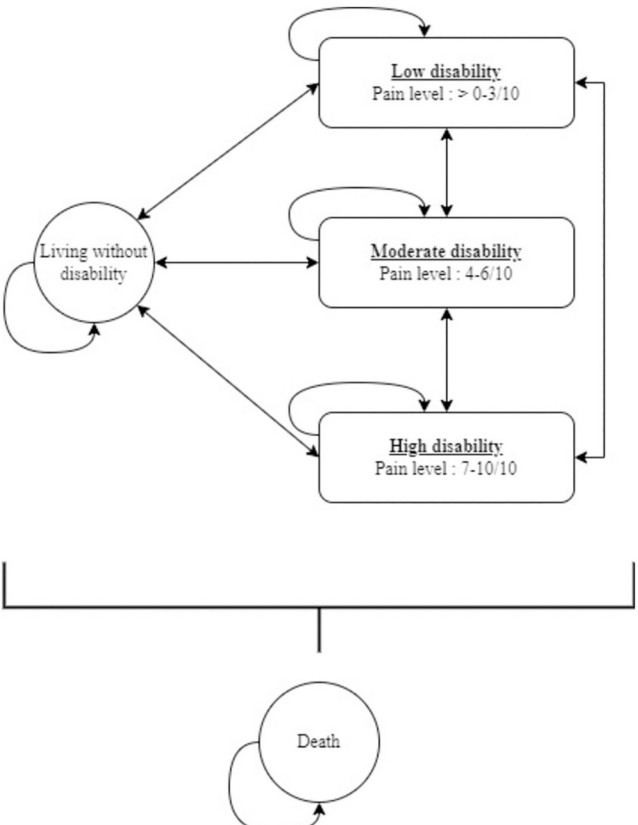

**Fig 3. Projected Markov model covering the period from three to 24 months post initial ED visit.**

influenced loss to follow-up at 1 and 3 months (e.g., socioeconomic status; level of pain and level of catastrophizing at the ED visit) will be examined to identify if they are indeed predictors of loss to follow-up. Characteristics shown to influence follow-up will be used within multiple imputations and within sensitivity analyses regarding these variables. Uncertainty in cost and effectiveness measures for the cost-utility analysis (Objective 2) will be obtained using non-parametric bootstrap resampling with replacement. Uncertainty in the hybrid model parameters (probabilities, costs, and efficiency) (Objective 3) will be obtained via a probabilistic sensitivity analysis performed using a Monte Carlo simulation. Three scenarios will be considered in the probabilistic sensitivity analyses. The first will assume that all participants with missing follow-up data in the control group will have utility scores at 3 months corresponding to those of the 95th percentile, while all those missing in the intervention group will have scores corresponding to those of the 5th percentile. For the second scenario, we will re-analyze all parameters included in the model, excluding outliers. Finally, in the third and last scenario, the different states of the Markov Model will be based not on the level of pain, but on the level of interference of pain on daily activities. Both uncertainties will be represented visually using a cost-effectiveness diagram, cost-effectiveness acceptability curve, and cost-effectiveness acceptability frontier [35]. As there are significant differences between men and women in the strategies used to cope with pain, and the use of health system services and resources during a painful episode (e.g., [56, 57]), subgroup analyses of men and women will be performed for both objectives (p < .05). Subgroup analyses will also be performed according to MSKD category (spine, upper extremity, and lower extremity) for both objectives (p < .05). All the statistical analyses carried out as part of this project and their detailed reporting in scientific publications will adhere to the CHEERS 2022 standard [58].

## Ethical considerations and data management

This project and all of its components (conduct of the RCT, three-month participants' follow-up, access to nominative and administrative databases) were approved by the CHU de Québec–UL's Ethical review board (approval number: MP-20-2019-4307). The randomized clinical trial was also registered with the US National Institutes of Health (#NCT04009369). Each participant signed a written consent form prior to participation. All data collected will be kept in a secure repository and destroyed thereafter. All members of the research team signed a confidentiality agreement.

## Discussion

The overall aim of this project is to evaluate the costs of different models of care for the management of MSKDs in the ED. This will be achieved through three specific objectives: 1) to compare the average costs of an ED consultation and care for various MSKDs; 2) to evaluate the incremental cost-effectiveness ratio (ICER) of two ED models of care for the management of MSKDs over a three-month period post-initial ED visit; and 3) to estimate the ICER between ED models of care for the management of MSKDs over a two-year period.

Until now, there has been no formal economic evaluation of the inclusion of a primary contact physiotherapist in the ED compared with usual practice (emergency physician). The only studies that have been done on the subject have assumed that the effectiveness of the primary contact physiotherapist's management in the ED is equivalent to that of usual care by the emergency physician. However, several studies have reported that primary contact physiotherapist management can reduce the use of services and resources during the ED stay [8, 26–29, 59]. This research project will fill an important need in the literature by providing an in-depth analysis of the costs and efficiency of the considered models of care. Indeed, this

project will help identify the most efficient ED model of care. These models of care also have the potential to improve the quality of services offered to people with MSKDs, their clinical evolution and their quality of life. The increased use of various health professionals in the management of patients in contexts such as the ED can pave the way for the development of new avenues of practice and potentially more efficient organization of services that will benefit the population.

This study is associated with some potential limitations. First of all, the data needed to carry out Objectives 1 and 2 will partly be based on results from a pilot pragmatic RCT. Therefore, the results obtained should be interpreted with caution. The small sample size (n = 78) could possibly limit analyses on the number of plausible branches in the final decision tree as well as the amount of subgroup analysis that will be performed. In addition, although high, the retention rate at the 3-month follow-up of the RCT was 80% [29], which implies that some data related to the use of services and resources, costs and health-related quality of life will be missing. However, this limitation will be mitigated using multiple imputation methods [55]. Sensitivity analyses will also be performed to assess the robustness of the results obtained. Finally, it may be difficult to obtain some of the data on medium- and long-term costs and measures of effectiveness for the ED models of care studied in Objective 3 from the scientific literature. Nevertheless, estimates can be obtained by soliciting the opinions of experts in the fields of MSKD management and emergency medicine, as this method is regularly used in modeling [35].

As for knowledge translation, following the project, formal presentations will be made to all key stakeholders at the CHU (emergency physicians, physiotherapists, nurses, orderlies, patient representatives and administrators) on site or remotely to present the results of the study and discuss lessons learned and future avenues. The results of this project will also be shared with provincial stakeholders (professional associations, patient associations and governments). They will also be disseminated at national and international scientific conferences on economics, health services organization and emergency services. Four manuscripts will be published in peer-reviewed journals. If successful, this project will help guide economic evaluations for a large-scale, multi-center trial aiming to improve the management of people presenting with a MKSD in the ED.

## Acknowledgments

The authors would like to thank the following persons for their contributions: project participants, Antony Barabé, PT, physiotherapist at the Centre Hospitalier de l'Université Laval (CHUL), the entire team of managers at the *Direction des services multidisciplinaires* of the CHU de Québec–Université Laval (Marie-Christine Laroche, Catherine Van Neste, Marie-Claude Brodeur and Stéphane Tremblay) for their support throughout the implementation of the project and its realization.

## Author Contributions

**Conceptualization:** Rose Gagnon, Luc J. Hébert, Jason R. Guertin, Simon Berthelot, François Desmeules, Kadija Perreault.

**Funding acquisition:** Rose Gagnon, Luc J. Hébert, Jason R. Guertin, Kadija Perreault.

**Methodology:** Rose Gagnon, Luc J. Hébert, Jason R. Guertin, Simon Berthelot, François Desmeules, Kadija Perreault.

**Project administration:** Rose Gagnon, Luc J. Hébert, Jason R. Guertin, Kadija Perreault.

**Visualization:** Rose Gagnon, Luc J. Hébert, Jason R. Guertin, Kadija Perreault.

**Writing – original draft:** Rose Gagnon.

**Writing – review & editing:** Rose Gagnon, Luc J. Hébert, Jason R. Guertin, Simon Berthelot, François Desmeules, Kadija Perreault.

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
