## [Decision Letter · Decision Letter 0]

9 Jun 2023

PONE-D-22-27339Integration of primary contact physiotherapists in the emergency department for individuals presenting with minor musculoskeletal disorders: Protocol for an economic evaluationPLOS ONE

Dear Dr. Gagnon,

Thank you for submitting your manuscript to PLOS ONE. After careful consideration, we feel that it has merit but does not fully meet PLOS ONE’s publication criteria as it currently stands. Therefore, we invite you to submit a revised version of the manuscript that addresses the points raised during the review process.

We look forward to receiving your revised manuscript.

Kind regards,

Joshua Robert Zadro, PhD

Academic Editor

PLOS ONE

Journal Requirements:

2. We note that you will be accessing administrative databases for data collection for your study. Please clarify which databases will be accessed and whether you have obtained the necessary ethics approval or permission to access these databases. Furthermore, please clarify whether these databases are anonymized.

“Part of the data that will be used in this study were acquired during a project that was supported by the CHU de Québec – Université Laval, subsidies from LJH and KP and a clinical research scholarship awarded to one of the CHU de Québec collaborators by the Fondation du CHU de Québec for the multidisciplinary council of the CHU de Québec – Université Laval . RG received scholarships from the Canadian Institute of Health Research (CIHR; https://cihr-irsc.gc.ca/e/193.html), the Fonds de recherche du Québec – Santé (FRQ-S; https://frq.gouv.qc.ca/sante/), the Unité de Soutien SSA – Québec (https://ssaquebec.ca/), the Ordre professionnel de la physiothérapie du Québec (OPPQ; https://oppq.qc.ca/), the Centre interdisciplinaire de recherche en réadaptation et intégration sociale (Cirris; https://www.cirris.ulaval.ca/) and Université Laval (https://www.fmed.ulaval.ca/). JRG, SB and FD are FRQ-S Research Scholars. The funders had and will not have a role in study design, data collection and analysis, decision to publish, or preparation of the manuscript.”

Additional Editor Comments (if provided):

Thank you for your patience Rose and apologies again for the delay. The reviewers are generally positive about the protocol but there is more detailed needed in numerous places. Please pay careful attention to the reviewers comments when revising your paper.

Reviewers' comments:

Reviewer's Responses to Questions

**Comments to the Author**

1. Does the manuscript provide a valid rationale for the proposed study, with clearly identified and justified research questions?

Reviewer #1: Yes

Reviewer #2: Yes

Reviewer #3: Yes

2. Is the protocol technically sound and planned in a manner that will lead to a meaningful outcome and allow testing the stated hypotheses?

Reviewer #1: Yes

Reviewer #2: Yes

Reviewer #3: Partly

3. Is the methodology feasible and described in sufficient detail to allow the work to be replicable?

Reviewer #1: Yes

Reviewer #2: Yes

Reviewer #3: Yes

4. Have the authors described where all data underlying the findings will be made available when the study is complete?

Reviewer #1: Yes

Reviewer #2: Yes

Reviewer #3: Yes

5. Is the manuscript presented in an intelligible fashion and written in standard English?

Reviewer #1: Yes

Reviewer #2: Yes

Reviewer #3: Yes

6. Review Comments to the Author

You may also provide optional suggestions and comments to authors that they might find helpful in planning their study.

Reviewer #1: This work has three objectives: to compare the average cost, to evaluate the ICER, and to estimate the ICER of the models of interest, which would be the combinations of physiotherapist and ED physician, ED physician alone, or physiotherapist alone.

1. Why only objective 3 consider the model with the group of primary contact physiotherapist management alone?

2. Authors propose to use decision tree for objective 2. However, tree is well-known to be overfitting. Please provide the information how this will be avoided in the analysis plan.

3. Authors argued to impute data using the Missing not at random approach. Please be specific how this will be implemented, especially if it’s not at random, what assumptions the authors will make? And whether it is reasonable in practice?

Reviewer #2: Just clarify if the total sample size is 78 or as read in lign 49-50 that data are collected during a randomized clinical trial (n=78) as well from CHU Québec administrative databases .Are they the same

Do you have an estimate of the number of men and women for the subgroup analyses.What about the lack of power if a group is too small?

Reviewer #3: Thank you for the opportunity to review this interesting work examining the cost-effectiveness of an physiotherapy led intervention in the emergency department setting. The ability to conduct an economic evaluation using clinical trial data associated with an allied healthcare intervention in the emergency department is not that common in the literature as outlined in the introduction of the paper. The structure of the paper is well organized however further clarification is required and along with the provision of more details about the methods prior to being suitable for publication.

Major comments

1. Although the manuscript describes a protocol for an economic evaluation, it would be of benefit to follow the CHEERS 2022 reporting guidance for economic evaluations. This is primarily applicable to the items outlined in the checklist associated with reporting of Methods (items 4-21).

Husereau, D., Drummond, M., Augustovski, F., et al. & CHEERS 2022 ISPOR Good Research Practices Task Force (2022). Consolidated Health Economic Evaluation Reporting Standards 2022 (CHEERS 2022) statement: updated reporting guidance for health economic evaluations. BMJ (Clinical research ed.), 376, e067975. https://doi.org/10.1136/bmj-2021-067975

2. Further description in the methods is needed to outline how the costing will be completed associated with the Recommendation and use of services and resources at baseline and 1- and 3-month follow-up visits as outlined in Table 3 of your published clinical trial paper pg. 854. How will uncertainty associated with these costs be addressed in the model? Also from Table 3 are any of the consultation with other professional further physiotherapy appointments with outpatient public or private healthcare providers? How will these visits be costed out?

3. With respect to Objective 3 in the manuscript please expand on the rationale for considering how a short-term ED physiotherapist intervention is to have a clinical impact and effect an individual’s healthcare resource utilization and costs over a two-year period. It is stated in the manuscript that follow-up for studies with similar interventions have had follow-up periods up to 2 years. What are some of the findings of these papers and their limitations that need to be considered associated with your utilization of a 2-year time horizon for the economic evaluation.

4. Please justify further why only a societal perspective was used for the analysis. It would seem to be important for this evaluation to provide the perspective of the health system (i.e. the hospital and the provincial payers) as well as just the patient perspective for out of pocket expenses and potential time off work.

5. Please provide further details regarding which specific CHU de Quebec – UL administrative dataases will be used and which public databases will be used. Do you have permission for the study to use hospital level administrative inpatient and outpatient reporting systems?

6. Further details regarding the calculation of dis-utility from baseline to 1 month and 3 months is required. It states in the manuscript that mean-utility scores will be used. Will you be determining the utility values using Area Under the Curve analysis for each patient? Using just the calculated mean values at each time point may not reflect the within patient changes in quality of life.

Minor comments

1. Please provide updated Figures 2 and 3. They did not upload appropriately in the pdf image and are not readable.

2. What is the cycle length associated with the Markov Model?

3. Further description of the anticipated sensitivity analyses.

4. Consider discussing the challenges faced so far with the development of the protocol and analysis plan.

7. PLOS authors have the option to publish the peer review history of their article (what does this mean?). If published, this will include your full peer review and any attached files.

Reviewer #1: No

Reviewer #2: No

Reviewer #3: **Yes: **James Bowen

---

## [Author Response · Author response to Decision Letter 0]

21 Jul 2023

Manuscript ID PONE-D-22-27339 / Plos One

“Integration of primary contact physiotherapists in the emergency department for individuals presenting with minor musculoskeletal disorders: Protocol for an economic evaluation”

POINT-BY-POINT RESPONSE TO THE REVIEWERS’ COMMENTS

We very much appreciate the reviewers’ comments and suggestions, which certainly have helped us improve the manuscript.

Editor’s comments to the Author

Please ensure that your manuscript meets PLOS ONE's style requirements, including those for file naming. The PLOS ONE style templates can be found at https://journals.plos.org/plosone/s/file?id=wjVg/PLOSOne_formatting_sample_main_body.pdf and https://journals.plos.org/plosone/s/file?id=ba62/PLOSOne_formatting_sample_title_authors_affiliations.pdf.

Author’s response:

Thank you for bringing this to our attention. We made the following changes to ensure that the manuscript meets PLOS ONE’s style requirements:

- Removal of the capitalized words in the title and subtitle

- Removal of the abbreviations in the affiliations

- Removal of all bold words throughout the abstract and the text

- Moving of the references before the punctuation marks

We note that you will be accessing administrative databases for data collection for your study. Please clarify which databases will be accessed and whether you have obtained the necessary ethics approval or permission to access these databases. Furthermore, please clarify whether these databases are anonymized.

Author’s response:

Details have been added to the abstract and the body of the manuscript (Pages 9-10, Lines 198-207) on the various CHU de Québec - Université Laval databases that will be used. Details have also been added as to whether these databases are nominative or not. Finally, the "Ethical considerations and data management" subsection has been modified and now begins as follows: “This project and all of its components (conduct of the RCT, three-month participants’ follow-up, access to nominative and administrative databases) were approved by…” (Page 15, Lines 333-335).

Thank you for stating in your Funding Statement:

“Part of the data that will be used in this study were acquired during a project that was supported by the CHU de Québec – Université Laval, subsidies from LJH and KP and a clinical research scholarship awarded to one of the CHU de Québec collaborators by the Fondation du CHU de Québec for the multidisciplinary council of the CHU de Québec – Université Laval . RG received scholarships from the Canadian Institute of Health Research (CIHR; https://cihr-irsc.gc.ca/e/193.html), the Fonds de recherche du Québec – Santé (FRQ-S; https://frq.gouv.qc.ca/sante/), the Unité de Soutien SSA – Québec (https://ssaquebec.ca/), the Ordre professionnel de la physiothérapie du Québec (OPPQ; https://oppq.qc.ca/), the Centre interdisciplinaire de recherche en réadaptation et intégration sociale (Cirris; https://www.cirris.ulaval.ca/) and Université Laval (https://www.fmed.ulaval.ca/). JRG, SB and FD are FRQ-S Research Scholars. The funders had and will not have a role in study design, data collection and analysis, decision to publish, or preparation of the manuscript.”

Author’s response:

Thank you for your comment. We included a modified funding statement in the cover letter. We hopefully provided clearer information regarding funding sources. As the sources of funding received are nominative and/or are not grants received from funding agencies, they do not have a specific grant number, except for the CIHR scholarship.

We note that you have stated that you will provide repository information for your data at acceptance. Should your manuscript be accepted for publication, we will hold it until you provide the relevant accession numbers or DOIs necessary to access your data. If you wish to make changes to your Data Availability statement, please describe these changes in your cover letter and we will update your Data Availability statement to reflect the information you provide.

Author’s response:

As this manuscript is a Study protocol, this clause does not apply. This is what is stated in the manuscript’s data availability statement.

Thank you for your patience Rose and apologies again for the delay. The reviewers are generally positive about the protocol but there is more detailed needed in numerous places. Please pay careful attention to the reviewers’ comments when revising your paper.

Author’s response:

Thank you to the Editor for working so hard to provide us with a quality peer review process. We have carefully read the reviewers' comments and used them to make significant changes to the manuscript.

Reviewer 1

This work has three objectives: to compare the average cost, to evaluate the ICER, and to estimate the ICER of the models of interest, which would be the combinations of physiotherapist and ED physician, ED physician alone, or physiotherapist alone.

Why only objective 3 consider the model with the group of primary contact physiotherapist management alone?

Author’s response:

Thank you for this relevant question. It is true that this choice is not clearly explained in the manuscript. We have added clarification to this effect in the "Data collection" subsection of the "Materials and Methods" section of the manuscript. The text now reads as follows: “Of note, the study population for the model of care consisting of primary contact physiotherapist management and discharge from the ED was not observed at all during the pilot pragmatic RCT. Consequently, this model of care will have to be modeled, and therefore cannot be studied in the context of Objectives 1 and 2, which are based on RCT data. All the parameters needed to represent this care model within Objective 3…” (Page 10, Lines 208-214).

Authors propose to use decision tree for objective 2. However, tree is well-known to be overfitting. Please provide the information how this will be avoided in the analysis plan.

Author’s response:

Thank you for highlighting this point with which we agree. Details have been added to the analysis plan, which now reads as follows: “The decision tree (Objective 3) will be created to reflect the results of the RCT as closely as possible and will therefore include all possible interventions and services used by a participant following the initial visit to the ED for each model of care considered (Fig 2). The pruning of each terminal node containing less than 5% of the participants will be determined based on discussions with a panel of experts. The conditional…” (Page 13, Lines 283-289)

Authors argued to impute data using the Missing not at random approach. Please be specific how this will be implemented, especially if it’s not at random, what assumptions the authors will make? And whether it is reasonable in practice?

Author’s response:

Thank you for your relevant comment. Based on the results obtained in the pilot randomized clinical trial, it will be assumed that some characteristics, such as those listed below, may put participants at greater risk of having been lost to follow-up at 1 and 3 months:

- Low socioeconomic status

- Level of pain at the time of the emergency visit

- Level of catastrophizing related to pain experience at ED visit.

Missing follow-up data will therefore be imputed using the Missing not at random technique, and sensitivity analyses will be performed on the chosen hypotheses. All necessary variables have already been collected during the randomized clinical trial.

Following this comment, details on the assumptions used have been added to the manuscript on Page 14, Lines 306 to 311.

Reviewer 2

Just clarify if the total sample size is 78 or as read in lign 49-50 that data are collected during a randomized clinical trial (n=78) as well from CHU Québec administrative databases. Are they the same

Author’s response:

We apologize for any confusion. Details have been added in the abstract on lines 51 and 52, as well as in the manuscript through the "Data collection" sub-section of the "Materials and Methods" section.

Do you have an estimate of the number of men and women for the subgroup analyses. What about the lack of power if a group is too small?

Author’s response: 

We would like to thank the reviewer for this interesting comment. 

We would like to note to the reviewer that statistical power considerations are not routinely considered within economic evaluations as their power depends, at least in part, on the willingness-to-pay threshold of the analysis. As this parameter is considered to be undefined, at least in Canada, the study’s power will also be undefined [Textbook of Pharmacoepidemiology, 2006].

Nonetheless, in Canada, the organization responsible for overseeing economic evaluation and issuing the various guidelines informing the work of health economists is the Canadian Agency for Drugs and Technologies in Health (CADTH). In their latest guidelines, issued in 2017, they state:

“The economic evaluation should reflect the entire target population as defined by the decision problem. Researchers should, however, examine any potential sources of heterogeneity that may lead to differences in parameter-input values across distinct subgroups. Note that heterogeneity may result from differences in the natural history of the disease, effectiveness of the interventions, health state preferences, or costs of the interventions. Heterogeneity may result in different decisions with respect to cost-effectiveness among different subgroups. The responsibility of the researcher, therefore, is to establish whether important heterogeneity exits in parameter estimates. A stratified analysis will allow decision-makers to identify any differential results across subgroups.”

[Guidelines for the Economic Evaluation of Health Technologies: Canada – 4th Edition, March 2017]

Considering the potential differences between men and women in the physiological mechanisms of pain, in the experience of pain and in the use of health system services and resources in the context of pain [e.g., Fullerton et al., 2018; Mills et al., 2019], it is important to carry out sub-group analyses based on sex and to report these results.

Reviewer 3

Thank you for the opportunity to review this interesting work examining the cost-effectiveness of a physiotherapy led intervention in the emergency department setting. The ability to conduct an economic evaluation using clinical trial data associated with an allied healthcare intervention in the emergency department is not that common in the literature as outlined in the introduction of the paper. The structure of the paper is well organized however further clarification is required and along with the provision of more details about the methods prior to being suitable for publication.

Although the manuscript describes a protocol for an economic evaluation, it would be of benefit to follow the CHEERS 2022 reporting guidance for economic evaluations. This is primarily applicable to the items outlined in the checklist associated with reporting of Methods (items 4-21).

Husereau, D., Drummond, M., Augustovski, F., et al. & CHEERS 2022 ISPOR Good Research Practices Task Force (2022). Consolidated Health Economic Evaluation Reporting Standards 2022 (CHEERS 2022) statement: updated reporting guidance for health economic evaluations. BMJ (Clinical research ed.), 376, e067975. https://doi.org/10.1136/bmj-2021-067975

Author’s response:

We thank the reviewer for his very pertinent comment. Details have been added in the "Data analysis and interpretation of results" sub-section of the "Materials and Methods" section of the manuscript. The text now reads as follows: “All the statistical analyses carried out as part of this project and their detailed reporting in scientific publications will adhere to the CHEERS 2022 standard [58].” (Page 15, Lines 329-331)

Further description in the methods is needed to outline how the costing will be completed associated with the Recommendation and use of services and resources at baseline and 1- and 3-month follow-up visits as outlined in Table 3 of your published clinical trial paper pg. 854. How will uncertainty associated with these costs be addressed in the model? Also from Table 3 are any of the consultation with other professional further physiotherapy appointments with outpatient public or private healthcare providers? How will these visits be costed out?

Author’s response :

The reviewer makes important points that deserved further clarification in the manuscript. The "Study outcomes" sub-section of the "Materials and Methods" section has been substantially modified to reflect the essence of this commentary. The changes can be found on Pages 11-12, Lines 228 to 256.

With respect to Objective 3 in the manuscript please expand on the rationale for considering how a short-term ED physiotherapist intervention is to have a clinical impact and effect an individual’s healthcare resource utilization and costs over a two-year period. It is stated in the manuscript that follow-up for studies with similar interventions have had follow-up periods up to 2 years. What are some of the findings of these papers and their limitations that need to be considered associated with your utilization of a 2-year time horizon for the economic evaluation.

Author’s response:

Thank you for this comment, which we believe deserves a detailed response as found below. The reviewer will find further explanations between lines 88 and 124 of the manuscript.

To start, between 25 to 30% of all emergency department visits are for musculoskeletal disorders. A significant proportion of people with a minor musculoskeletal disorder will present pain related to this disorder more than a year after its onset. Currently, 18.9% of the Canadian population suffers from persistent pain [Schopflocher et al., 2011]. The presence of persistent pain is associated with greater use of health system services and resources, such as medication and consultations with various health professionals. Hence, the impacts of such musculoskeletal disorders can manifest in the long term and require assessment in the long term.

Also, in response to the reviewer’s comment, previous research supports the potential for long term benefits of a brief physiotherapy intervention. Indeed, early treatment by a physiotherapist is associated with a reduction in pain levels, psychological symptoms and the risk of developing persistent pain. Other studies have reported that early management by a physiotherapist in a primary care setting is also associated with a reduction in costs and a slight increase in health-related quality of life up to two years after initial treatment compared to usual management by a family physician [Bornhöft et al, 2019; Denninger et al, 2017]. However, no study has measured the long-term effects of this kind of management in the ED. 

The results of our previous RCT [Gagnon et al, 2021] support the favorable clinical outcomes of the integration of physiotherapy in the ED compared to usual medical practice 3 months after the ED visit. Analysing outcomes up to two years will allow to provide new results to inform the implementation of ED models for the management of musculoskeletal disorders.

Please justify further why only a societal perspective was used for the analysis. It would seem to be important for this evaluation to provide the perspective of the health system (i.e. the hospital and the provincial payers) as well as just the patient perspective for out of pocket expenses and potential time off work.

Author’s response:

We thank the reviewer for his comments, with which we agree. The abstract and manuscript have been modified to reflect that we will be studying two different perspectives, the Health system perspective and the Societal perspective.

Please provide further details regarding which specific CHU de Quebec – UL administrative dataases will be used and which public databases will be used. Do you have permission for the study to use hospital level administrative inpatient and outpatient reporting systems?

Author’s response:

This comment has already been addressed previously. The reviewer will be able to find the information on the databases and Ethics approval on Lines 198 to 207 and 333-335.

Further details regarding the calculation of dis-utility from baseline to 1 month and 3 months is required. It states in the manuscript that mean-utility scores will be used. Will you be determining the utility values using Area Under the Curve analysis for each patient? Using just the calculated mean values at each time point may not reflect the within patient changes in quality of life.

Author’s response:

The requested clarifications have been added to the manuscript, which now reads as follows: “The difference in utility scores between the 3-month follow-up and baseline will be calculated for each participant using area-under-the-curve analyses. Once the difference in utility scores will be calculated for each participant, the differences will be averaged to obtain the average gain or loss in utility scores per ED care model. The mean gain or loss in utility scores will then be transformed into QALYs. The efficiency values…” (Page 12, Lines 260-264).

Please provide updated Figures 2 and 3. They did not upload appropriately in the pdf image and are not readable.

Author’s response:

We apologize for the inconvenience. Figures 2 and 3 have been updated to appear correctly on the journal submission platform.

What is the cycle length associated with the Markov Model?

Author's response:

This precision can be found in the manuscript on Page 8, Lines 170 to 172. The text reads as follows: “Thus, the Markov model will cover a 24-month period. It will include two-week cycles in order to capture the clinical evolution of the patients included.”

Further description of the anticipated sensitivity analyses.

Author's response:

We thank the reviewer for his relevant comment and have added the requested specifications to the "Data analysis and interpretation of results" sub-section of the "Materials and Methods" section. The text now reads as follows: “Uncertainty in the hybrid model parameters (probabilities, costs, and efficiency) (Objective 3) will be obtained via a probabilistic sensitivity analysis performed using a Monte Carlo simulation. Three scenarios will be considered in the probabilistic sensitivity analyses. The first will assume that all participants with missing follow-up data in the control group will have utility scores at 3 months corresponding to those of the 95th percentile, while all those missing in the intervention group will have scores corresponding to those of the 5th percentile. For the the second scenario, we will re-analyze all parameters included in the model, excluding outliers. Finally, in the third and last scenario, the different states of the Markov Model will be based not on the level of pain, but on the level of interference of pain on daily activities. Both uncertainties…” (Page 14 and 15, Lines 313-322).

Consider discussing the challenges faced so far with the development of the protocol and analysis plan.

Author's response:

We thank the reviewer for his insightful comment. This suggestion is highly appropriate, but it might be premature to initiate this discussion at this stage of the project. The project's post-mortem will enable us to identify the challenges encountered and the lessons learned which will be discussed thoroughly in the four manuscripts that we intend to publish in peer-reviewed journals, as well as in the first author’s doctoral thesis.

---

## [Decision Letter · Decision Letter 1]

29 Aug 2023

Integration of primary contact physiotherapists in the emergency department for individuals presenting with minor musculoskeletal disorders: protocol for an economic evaluation

PONE-D-22-27339R1

Dear Dr. Gagnon,

We’re pleased to inform you that your manuscript has been judged scientifically suitable for publication and will be formally accepted for publication once it meets all outstanding technical requirements.

Kind regards,

Joshua Robert Zadro, PhD

Academic Editor

PLOS ONE

Additional Editor Comments (optional):

Reviewers' comments:

Reviewer's Responses to Questions

**Comments to the Author**

1. Does the manuscript provide a valid rationale for the proposed study, with clearly identified and justified research questions?

Reviewer #1: Yes

Reviewer #3: Yes

2. Is the protocol technically sound and planned in a manner that will lead to a meaningful outcome and allow testing the stated hypotheses?

Reviewer #1: Yes

Reviewer #3: Yes

3. Is the methodology feasible and described in sufficient detail to allow the work to be replicable?

Reviewer #1: Yes

Reviewer #3: Yes

4. Have the authors described where all data underlying the findings will be made available when the study is complete?

Reviewer #1: Yes

Reviewer #3: Yes

5. Is the manuscript presented in an intelligible fashion and written in standard English?

Reviewer #1: Yes

Reviewer #3: Yes

6. Review Comments to the Author

You may also provide optional suggestions and comments to authors that they might find helpful in planning their study.

Reviewer #1: thanks for the response. all comments have been successfully addressed and I have no further comments.

Reviewer #3: Thank you for addressing the comments of the reviewers. The methodology associated with the economic evaluation are more fully described in the revised manuscript.

7. PLOS authors have the option to publish the peer review history of their article (what does this mean?). If published, this will include your full peer review and any attached files.

Reviewer #1: No

Reviewer #3: **Yes: **James M. Bowen

---

## [Editor Report · Acceptance letter]

1 Sep 2023

PONE-D-22-27339R1 

Integration of primary contact physiotherapists in the emergency department for individuals presenting with minor musculoskeletal disorders: protocol for an economic evaluation 

Dear Dr. Gagnon:

I'm pleased to inform you that your manuscript has been deemed suitable for publication in PLOS ONE. Congratulations! Your manuscript is now with our production department. 

Kind regards, 

on behalf of

Dr. Joshua Robert Zadro 

Academic Editor

PLOS ONE